# Nutrition Profile of Products with Cartoon Animations on the Packaging: A UK Cross-Sectional Survey of Foods and Drinks

**DOI:** 10.3390/nu12030707

**Published:** 2020-03-06

**Authors:** Sonia Pombo-Rodrigues, Kawther M. Hashem, Monique Tan, Zoe Davies, Feng J. He, Graham A. MacGregor

**Affiliations:** Wolfson Institute of Preventive Medicine, Barts and The London School of Medicine & Dentistry, Queen Mary University of London, EC1M 6BQ London, UK; k.hashem@qmul.ac.uk (K.M.H.); m.tan@qmul.ac.uk (M.T.); z.davies@qmul.ac.uk (Z.D.); f.he@qmul.ac.uk (F.J.H.); g.macgregor@qmul.ac.uk (G.A.M.)

**Keywords:** salt, fat, sugar, nutrient profiling model, labelling, cartoons, HFSS, food marketing

## Abstract

Background: Marketing, including the use of cartoon animations on packaging, has been shown to influence the food children choose to eat. This paper aims to determine the nutritional quality of UK food and drink products featuring child-friendly characters on pack. Methods: A comprehensive cross-sectional survey of food and drink with packaging appealing to children available in the UK. Products were classified high in fat, salt and/or sugar (HFSS) according to the UK nutrient profiling model and guidance for front of pack nutrition labelling. Logistic regression was used to determine whether there was a significant relationship between nutritional quality of products, and animation type. Results: Over half (51%) of 532 products with animations on packaging were classified as HFSS. Food products featuring unlicensed characters were significantly more likely to be deemed HFSS than those with licensed characters, according to both the nutrient profiling model (odds ratio (OR) 2.1, 95% CI: 1.3 to 3.4) and front of pack nutrition labelling system (OR 2.3, 95% confidence interval CI: 1.4 to 3.7). Conclusions: The use of cartoon characters on HFSS products is widespread. Policies to restrict the use of such marketing tactics should be considered to prevent children being targeted with unhealthy foods and drinks.

## 1. Introduction

Currently, children consume excess amounts of calories, fat, saturated fat, salt and sugar. The latest set of comprehensive data on diet and nutrition intake in the UK population from 2015/2016 [1] found children exceed the government recommendation on free sugars by an average of 8.3% (total average intake of 13.3% for those between the ages of 1.5 years and 18 years). Those between the ages of 4 and 18 were also exceeding an adult woman’s recommendation of consuming no more than 20 g of saturated fat a day. In addition to this, children are also not meeting key recommendations on nutrients more beneficial for their health, including fibre, fruit and vegetable intake. Between the ages of 1.5 and 18 years, an average of just 7.6% children were meeting their fibre recommendation, lower than in 2008/2009. Children aged 11–18 years consumed just 2.7 portions a day of fruit and vegetables, with only 8% meeting the recommended five a day target.

Research shows that marketing greatly influences the food children choose to eat [2,3], with children as young as three showing a preference for branded foods over identical unbranded products [4]. Cartoon animations on packaging is one form of marketing that the food and drink industry uses to influence children [5]. Animations range from the use of licensed TV, book and film characters such as Peppa Pig and Roald Dhal characters, to unlicensed characters created by the manufacturers themselves. The power of animation on advertising appears to be strongest when the character is better recognised. A study in New York showed that more popular characters increased apple intake in schools among children [6]. However, if the characters were unfamiliar they had no effect on apple uptake. This is also apparent in the UK, with a recent study showing children were significantly more likely to prefer food with familiar popular characters [7]. These findings are broadly aligned with those of a systematic review, which found that use of cartoon characters in marketing can positively increase children’s fruit or vegetable intake. Familiar licensed characters in particular, can have a more powerful influence on children’s food preferences, choices and intake, especially for energy-dense and nutrient-poor foods compared with fruits or vegetables [8].

Despite the emerging evidence demonstrating the potential influence of such marketing tactics on children’s food preference, there has been limited policy to its use towards healthier products. In the UK, some restrictions exist to limit advertising of ‘unhealthy’ food and drink to children. The Advertising Code set by the Committee for Advertising Practices (CAP/BCAP) [9] does not allow broadcasting of adverts for products classified as HFSS, using the Department of Health & Social Care’s (DHSC) nutrient profiling model, before, during or after programming with an audience of more than 25% children aged under 16 years. In 2019, under ground-breaking measures to tackle child obesity and following a public consultation, Transport for London began applying similar restrictions of HFSS food and drink advertising across its entire network [10]. In 2018, the UK government announced their intention to make more progress on reducing the marketing and promotion of HFSS food and drink [11], however marketing on packaging was excluded. The purpose of this study is to document the nutrition content of products with packaging that would appeal to young children available in all major UK retailers, and establish if they would be considered unhealthy according to two models; DHSC’s nutrient profiling model, which is widely used by industry, and the UK’s front of pack colour coded labelling system.

## 2. Materials & Methods

The data was collected from nutrition information panels on product packaging in 2019. The survey was designed as a comprehensive survey of all products with packaging that may appeal to children available in a snapshot in time.

### 2.1. Data Collection

For each product, the data collected included the company name, product name, pack weight, serving size and full nutrition information per 100g and per portion, if provided. All data were double-checked after entry, and a further 5% of entries were checked against the original source in a random selection of products. Under EU labelling legislation, nutritional information for products where a nutrient is negligible is allowed to be labelled as ‘trace’ or provided with ‘<’. In these circumstances, where any nutrient was displayed as ‘trace’, this was replaced with 0. Similarly where the nutrient content was <0.01, this was replaced with 0.01, <0.1 was replaced with 0.1, and <0.25 was replaced with 0.25 [12].

### 2.2. Stores

Data were collected in one large outlet in London from each of the major UK supermarkets (Aldi, Asda, Lidl, Iceland, Marks and Spencer, Morrisons, Sainsbury’s, Tesco, The Co-operative and Waitrose) as these supermarkets collectively hold 94.8% of the grocery market share [13].

### 2.3. Inclusion/Exclusion Criteria

Duplicates of the same product with different packaging sizes were excluded. Inclusion criteria of what would constitute as ‘child-friendly’ packaging are:Products with animated brand equity/licensed cartoon characters on products, as well as non-animated TV shows aimed at children (e.g., Mr Tumble, Teletubbies)Products with an animated character on front of pack.

Exclusion criteria:Products where the character is embedded in the manufacturer’s logo (e.g., Pringles, Laughing Cow)Seasonal products (e.g., Easter Eggs)Large occasional/celebration cakesMissing nutrition information

### 2.4. Nutritional Assessment of Products

*Per 100g:* Some brands sell the same formulation in different serving sizes. Therefore, the 100g data only included an example of one formulation regardless of the different serving sizes.

*Per serving:* The per serving data included all the different per serving and per portion available, or pack size.

*Nutrient Profiling Model*: Products were classified as HFSS using the DHSC nutrient profiling model [14]; if foods score 4 or more points and drinks score 1 or more, they would be deemed HFSS.

*High, Medium and Low criteria for sugars content:* The sugars content of food and drink surveyed was compared to the UK front of pack colour-coded labelling [15]. Colour coding is based on the following criteria for food (red/high >27 g/portion or >22.5 g/100 g, amber/medium >5.0 and ≤22.5 g/100 g, green/low ≤5.0 g/100 g) and drink (red/high >13.5 g/portion or >11.25 g/100 mL, amber/medium >2.5 and ≤11.25g/100 mL, green/low ≤2.5 g/100mL).

*High, Medium and Low criteria for salt content:* The salt content of food surveyed was compared to the UK front of pack colour-coded labelling. Colour coding is based on the following criteria: red/high >1.8 g/portion or >1.5 g/100 g, amber/medium >0.3 and ≤1.5 g/100 g, green/low ≤0.3 g/100 g).

*High, Medium and Low criteria for fat content:* The fat content of food surveyed was compared to the UK front of pack colour-coded labelling. Colour coding is based on the following criteria: red/high >21 g/portion or >17.5 g/100 g, amber/medium >3.0 and ≤17.5 g/100 g, green/low ≤3.0 g/100 g).

*High, Medium and Low criteria for saturated fat content:* The saturated fat content of food surveyed was compared to the UK front of pack colour-coded labelling. Colour coding is based on the following: red/high >6.0 g/portion or >5.0 g/100 g, amber/medium >1.5 and ≤5.0 g/100 g, green/low ≤1.5 g/100 g).

### 2.5. Statistical Analyses

The number and proportion of products classified as HFSS (according to the nutrient profiling model and the front of pack colour-coded labelling system) were calculated for all food and drink, and separately for products with licensed vs unlicensed characters on pack.

Logistic regression was used to determine whether a relationship was present between nutritional assessment of food and drink (as determined by nutrient profiling model and front of pack colour coding criteria) and type of animation (i.e., licensed character vs unlicensed). For unprocessed products with no nutrition label (i.e., fruits, vegetables, plain water), energy, fat, saturated fat, sugars, non-starch polysaccharides fibre, protein and salt content were obtained from external validated sources [16]. A *p*-value of <0.05 was considered significant. All analyses were conducted using R version 3.5.1.

## 3. Results

Nutritional information were collected from 534 food and drink products with some form of animation on packaging. Of those, nine products (seven unprocessed fruit, vegetable, water; one chocolate product; one jelly dessert) did not provide nutrition information on pack. Nutritional data were imputed for the seven unprocessed goods and as such, 532 food and drink products were included in the analysis (Table 1).

56% (*n* = 295) of all products surveyed were from food categories that are not recommended for frequent consumption (e.g., biscuits, cakes, chocolate, desserts, snack bars, crisps, milkshakes, ice cream, sugar sweetened yogurts and sugar confectionery). 42% (*n* = 223) of all products surveyed were high in sugar, 5% (*n* = 26) were high in salt, 17% products (*n* = 87) were high in fat, 18% products (*n* = 97) were high in saturated fat.

### 3.1. Nutritional Assessment

Over half (51%, *n* = 269) of the products surveyed were high in fat, saturated fat, sugar and/or salt, as deemed by DHSC’s front of pack colour coded labelling criteria. Similarly, when using the nutrient profiling model, 51% (*n* = 272) would be deemed HFSS (Figure 1).

### 3.2. Licenced vs Unlicensed Characters

Of the products surveyed, 17% (*n* = 92) used licensed characters that are often well recognised by young children (e.g., Disney, Peppa Pig and Paw Patrol). Of these, 35% (*n* = 32) had a red label for either fat, saturated fat, sugars and/or salt, and 37% (*n* = 34) would have a score of ≥4 (≥1 for drink) and thus be classified as HFSS, under the nutrient profiling model.

The remainder of products (*n* = 440) used unlicensed characters and animations, of which 54% (*n* = 237) would have received a red warning label on pack for either fat, saturated fat, sugars and/or salt, and 54% (*n* = 238) would have a score of ≥4 (≥1 for drink) under the nutrient profiling model.

In foods, there was a significant relationship between license type and HFSS classification as defined by the nutrient profile score (*p* = 0.002) or the front of pack colour-coding guidelines (*p* < 0.001). In both classification methods, products featuring unlicensed characters were twice as likely to be deemed HFSS (odds ratio (OR) 2.1, 95% confidence interval (CI): 1.3 to 3.4 and OR 2.3, 95% CI: 1.4 to 3.7, respectively).

A small proportion (3%, *n* = 18) of products with cartoon animations on their packaging were found in unprocessed fruit, vegetables and plain water. The majority of these were unlicensed characters on retailer products, with only three products using better-known licensed characters e.g., Lego or Disney. Lidl came out as the retailer with the highest use of cartoon characters on fruits and vegetables as part of their Oaklands range of fruit and vegetables (11 products out of 15 from retailers).

## 4. Discussion

This comprehensive survey shows the current use of cartoon characters on packaging as a form of advertising by industry, to children. As evidenced from this research, a high percentage (51%) of products with child-friendly packaging would be classified as HFSS. This pattern is not just specific to the UK market, with research in different countries also documenting the extensive use of culturally tailored characters by the food and drink industry to market primarily energy dense and nutrient poor products to children in different settings [8]. This method of marketing is used to exploit children and influence their purchasing habits, by developing an emotional relationship and encouraging brand loyalty that persists into adulthood [17,18,19,20]. Additionally, companies use cartoon characters to promote their products to children so as to ultimately increase sales and maximise their market share, despite many of these products contributing to a poor diet [19].

Our findings also demonstrated a stronger use of unlicensed characters in food and drink packaging compared to licensed characters. This could be due to financial logistics, whereby food and drink companies using unlicensed characters are able to retain 100% of the revenues, whilst licensed characters are owned by entertainment companies and thus receive royalties in exchange for use of the character. Some licensed companies have also implemented nutrition guidelines as part of their corporate social responsibility, which restricts the use of their characters on food and drink, which do not meet their criteria. Disney, for example, was the first media company to develop nutrition guidelines, which set limits on fat, salt, sugar and portion sizes. They have also developed their own food and drink range (Disney Kitchen), signposting key nutritional information to parents [21].

The use of cartoon characters for advertising of HFSS food and drink is not a novel marketing strategy, with a similar study conducted 19 years ago demonstrating little improvement. Then, 77% contained high levels of sugar (>10 g per 100 g), salt (>1.25 g per 100 g), saturated fat (>20 g per 100 g) or total fat (5 g <100 g), even though soft drinks and confectionery were excluded, and just 7% of the products were low in fat (≤3 g per 100 g), saturated fat (≤1 g per 100 g), sugar (≤2 g per 100 g) and salt (≤0.25 g per 100 g) [5]. Whilst a direct comparison between these studies may not be appropriate, both highlight a large proportion of food and drink products high in fat, saturated fat, sugars and salt.

Companies continue to use unfavourable marketing tactics to influence children’s food and drink preferences that are unaligned with current dietary advice. As previously mentioned, strong evidence from systematic reviews has shown that the marketing of unhealthy food influences children’s preferences, purchasing habits (through pester power) and consumption patterns [3]. This has contributed to the current obesity epidemic, which is occurring on a global scale. In 2016, the World Health Organization (WHO) estimated there were 340 million overweight or obese children and adolescents aged 5–19 in the world; this equates to 18% girls and 19% boys globally [22]. In England and Wales, one third of children are obese or overweight by the age of 11 years, with around 6,000 young people being diagnosed with type 2 diabetes each year, and almost one quarter of 5 years olds having obvious tooth decay [23,24,25].

In an effort to combat childhood obesity and reshape our environment, policy-makers and researchers have proposed an array of measures to halt or reverse this trend, including reformulation programmes to reduce the total amount of salt, sugar and calories in food, and restrictions in advertising. However, although restrictions have been set for advertising of HFSS food and drink, these do not currently apply to advertising on food packaging, despite the influence it has been shown to have on purchasing decisions. If restrictions were extended to packaging, then 51% of products currently on the market would fail CAP’s eligibility criteria and therefore would not be allowed to be advertised during children’s programming.

There are a number of ways that companies can be encouraged to change their practices. As research would suggest, there is an opportunity for companies to promote healthier products with the use of animations [26,27]. Some retailers and manufacturers are doing this to some extent. For instance, in 2000 Sainsbury’s sold fruit packaged in wrappers with Looney Tunes cartoon characters and Iceland sold frozen vegetables such as baby corn and broccoli mix with cartoon characters on their packaging [5]. Some retailers are currently leading the way, with Lidl proclaiming to be the first British supermarket to introduce a range of fresh fruit and vegetables specifically targeted to children [28].

Another option could be that entertainment companies that license their characters for use on product packaging can create stricter nutrition criteria, which would restrict the use of their characters on unhealthy food and drink. Food and drink companies can also diversify their product portfolio so that animations only appear on non-HFSS food and drink. At the same time, they can reformulate their products to fall below HFSS thresholds.

From a government policy level, the UK government can introduce restrictions on the use of such cartoon characters—whether licensed or unlicensed—on HFSS product packaging. This could be an extension of current restrictions on the use of licensed characters and celebrities popular with children (under 16 years) in broadcast advertising on HFSS food and drink. Studies have demonstrated the power of children’s purchasing request behaviour, i.e., pester power, and how it strongly influences parents in making unhealthier choices [29,30,31]. Therefore, placing such restrictions could potentially have a positive effect on subsequent purchasing habits, as children are less likely to pester their parents into buying them.

Currently, the only country to have restricted the use of such marketing tactics on unhealthy products is Chile. In 2012, the Chilean government approved the Law of Nutritional Composition of Food and Advertising, which came into effect on 27 June 2016. The regulation defined certain limits for calories, saturated fat, sugar and salt content considered “high” in food and beverages. Products “high in” unhealthy nutrients must not be advertised directly to children under the age of 14 years, which includes the use of promotional strategies and incentives such as cartoons, animations and toys that could attract the attention of children [32].

## 5. Limitations

Our study was based on nutrition information provided on product packaging labels in store; therefore we have relied on the accuracy of the data provided on the label. It is assumed that the manufacturers provide accurate and up to date information in line with regulations. Plain unprocessed foods such as fruits and vegetables do not typically display nutritional information on pack in the UK, but in order to provide a fair representation of the market, we looked to external and validated sources of information to provide an exhaustive sample of products with animations on packaging.

We established our own inclusion and exclusion criteria; other researchers may include more products, including logos that are animated, which for the purposes of this study we have excluded. Our data did not include products out of stock on the day of collection, or products in the out of home sector, such as Happy Meals. Additionally, products with no animations on packaging were not included in the research, and as such, a reference group is lacking. Future studies should endeavor to include this type of data too, especially as purchases of these types of foods have increased in recent years and food eaten out of the home now accounts for a growing proportion of the total amount of food consumed, with more than a quarter of adults and one fifth of children in England buying and consuming food out of home/on the go at least once a week [33].

Nevertheless, the results of this study are relevant and serve to document the widespread use of child-friendly packaging on food and drink products sold in the UK, providing baseline data to evaluate public health interventions or policies that may influence such practices.

## 6. Conclusion

This analysis demonstrates that cartoon animation and characters, which appeal to children, are being used by food manufacturers and retailers to sell unhealthy foods which are high in fat, salt and/or sugar. The majority of food and drink featuring this type of packaging come from categories that would not be recommended for frequent consumption, such as cakes, biscuits and confectionary. Evidence has shown that cartoon animations on pack can influence eating habits in children. There is therefore an opportunity for the food industry to use this method of marketing in a more positive way, by promoting more fresh vegetables, fruit and water consumption. There is also opportunity for Government-led regulation to force companies to only use such marketing tactics to promote non-HFSS products.

## Figures and Tables

**Figure 1 nutrients-12-00707-f001:**
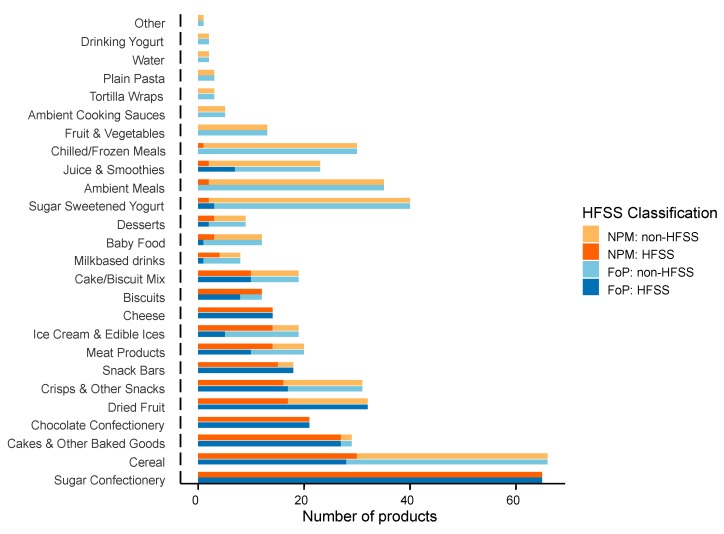
Summary of products by category, highlighting HFSS products as determined by the Nutrient Profiling Model (NPM) and UK Front of Pack nutrition labelling system (FoP).

**Table 1 nutrients-12-00707-t001:** Summary of food and drink products deemed HFSS as determined by the Nutrient Profiling Model (NPM) and UK Front of Pack nutrition labelling system (FoP).

Products	Total (n)	HFSS as Determined by NPM (%)	HFSS as Determined by FoP (%)	Belongs to Category not Recommended for Frequent Consumption * (%)
Overall	532	51.1	50.6	51.9
By license type:				
Licensed	92	37	34.8	34.8
Unlicensed	440	54.1	53.9	55.5
By product type:				
Food products	501	53.1	52.1	49.3
Drink products	31	19.4	25.8	93.5

* Food and drink categories that are not recommended for frequent consumption include biscuits, cake/biscuit mix, cakes and other baked goods, chocolate confectionery, crisps and other snacks, desserts, ice cream and edible ices, juice and smoothies, milk based drinks, other, snack bars, sugar confectionery and sugar sweetened yogurt.

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
