# Peer review of "Nutrition Profile of Products with Cartoon Animations on the Packaging: A UK Cross-Sectional Survey of Foods and Drinks"

_nutrients, 2020, doi:10.3390/nu12030707_

Round 1

Reviewer 1 Report

The paper presents an analysis of foods marketed to children and classifies these by their grading on a scale of "healthiness". The paper is generally well-written and presented, however, there were a number of issues I would like to discuss.

The manuscript follows the convention of nutrition of assuming that saturated fat and salt intake are harmful and 'unhealthy'. The authors may not be aware that there is considerable academic controversy about whether or not saturated fat is harmful, and also about salt. At least some of this controversy needs to be acknowledged, and I believe it would be useful to classify the products according to their sugar content only. This is at least, a component of food that is universally agreed to be harmful. As it stands, considering my different nutritional beliefs from that of the authors, the manuscript is of little interest to me as it mixes foods that I would consider harmful and those that are not.

About the figure, the food categories are ranked by the number of products.
I believe the figure would be more interpretable if it were ranked by the
the proportion of unhealthy products, with the category with the greatest
the proportion of unhealthy products uppermost.

Author Response

Many thanks for your feedback on our research. Please find below our responses to your comments:

1.The manuscript follows the convention of nutrition of assuming that saturated fat and salt intake are harmful and 'unhealthy'. The authors may not be aware that there is considerable academic controversy about whether or not saturated fat is harmful, and also about salt. At least some of this controversy needs to be acknowledged, and I believe it would be useful to classify the products according to their sugar content only. This is at least, a component of food that is universally agreed to be harmful. As it stands, considering my different nutritional beliefs from that of the authors, the manuscript is of little interest to me as it mixes foods that I would consider harmful and those that are not.

RESPONSE: The evidence addressing the health implications of excess salt intake on overall health, is overwhelming. This week alone a meta-analysis published in the British Medical Journal, provides new and strong evidence to support salt reduction as a key public health strategy to lower blood pressure and reduce the risk of strokes and heart disease. Whilst there is some research suggesting too little salt is also bad for health, the methodology has been questioned internationally, as described here (https://www.ahajournals.org/doi/10.1161/HYPERTENSIONAHA.119.13117). The debate around salt has been comprehensively reviewed and covered by several articles. As such, we have not added anything further on this to our current paper as we do not feel it is within the scope of the research in question.

With regards to saturated fat, a recent review published last year by the Scientific Advisory Committee on Nutrition (SACN) who advises Public Health England and other government organisations on nutrition and related health matters, looked at the effect of saturated fat on a number of health outcomes. Based on 47 systematic reviews and meta-analyses, the committee concluded that higher saturated fat consumption is linked to raised blood cholesterol and associated with increased risk of heart disease. Current recommendations in the UK are that we should be eating fewer foods high in saturated fat, and in smaller amounts.

https://assets.publishing.service.gov.uk/government/uploads/system/uploads/attachment_data/file/814995/SACN_report_on_saturated_fat_and_health.pdf

We feel it may not be appropriate to assess products solely on their sugar content. Nutrient profiling is a tool designed to help differentiate between foods and drinks that are more likely to be part of a healthy diet from those that are less likely (notably those foods that may contribute to excess consumption of energy, saturated fats, sugar or salt). Nutrient profiling models are used internationally when considering the implementation of restrictions on the marketing of foods to children, and is recognised by the World Health Organisation as a useful tool for a variety of applications. With this in mind, we have decided to look at the overall nutritional quality of a food or drink, and not just one isolated nutrient.

2. About the figure, the food categories are ranked by the number of products. 
I believe the figure would be more interpretable if it were ranked by the 
the proportion of unhealthy products, with the category with the greatest
the proportion of unhealthy products uppermost.

RESPONSE: We have amended the graph accordingly to reflect this. This has been included in the manuscript.

Reviewer 2 Report

This manuscript is presenting an analysis of the nutritional profiles of products aimed at children with a special focus on products using animated characters.

The study is scientifically sound and the results are of interest. The introduction includes sufficient information and the sampling techniques are clear and appropriate. 

Comments: Why were products with animated characters embedded in the logo excluded. According to the EU regulation logos are classified as health claims so I would guess as an extension of this one could consider animations in the logo to be the same as licensed characters. Especially in products like the Laughing cow with a clear positioning towards children. How many products were identified in this category? I would argue that they could be added to the analysis or if there are enough products to be analysed separately.

One issue with the current study is the lack of a reference group. Did you find any products targeted to children without any cartoons? Products clearly aimed to children? If yes do you have these data to add as a third category? If no could you maybe add a line in the results section to highlight this? It is an important finding in terms of marketing practices.

Lines 138-139: Instead of imputing data from food composition databases (I guess) I would suggest to directly classify those products as non HFSS and green in all traffic lights. Imputation should be carried out for the other two products either from a national food composition database or the average of similar products. However excluding those two products is acceptable and safer.

Lines 145: Please remove the word healthiness. The nutrient profiling system you are using (in fact any profiling system) does not measure healthiness. The concept of a healthy food is vague and scientifically complex. The nutrient profiling model used in the paper measures whether a product is HFSS and hence unsuitable to be marketed to children specifically on TV. In fact, you need to revisit the manuscript and remove any comment suggesting that a nutrient profiling model classifies a food as healthy or unhealthy... The systems you used classify as high in a given or multiple nutrients and suitable or not to be marketed to children. 

Lines 152: remove comment on unhealthy... cheese would be classified as red and you could hardily argue that cheese is unhealthy. Refrain from comments and interpretations in the results section

The sample for fruits and vegetables is too small to be analysed alone.

Incorporate it to the previous sections licenced and non licensed characters with a comment on how many of the products in each analysis were fruits & vegetables. 

Lines 191-196: You attempt a comparison with the previous survey but it is not possible for the reader to make the comparison. So if analysed altogether what are the proportion classified as HFSS. Doing the math quickly I would say approximately 50% of the total sample.. I would argue that this is an improvement from the 77%.

Author Response

Many thanks for reviewing our manuscript, and for your useful comments.

1. Why were products with animated characters embedded in the logo excluded.

RESPONSE: For the purpose this research we looked at the packaging of the food or drink in question and determined whether the packaging itself had animations that would likely be marketed specifically to children. We did not include animations within logo’s as it could be argued that they are static images across their whole portfolio of food and drink, some of which would not necessarily be marketed to children e.g. pringles, or homepride. These products were not collected and therefore unfortunately could not be included in the research. But thank you for your input on this – we will take this into consideration for future research.

2. One issue with the current study is the lack of a reference group. Did you find any products targeted to children without any cartoons? 

RESPONSE: Unfortunately we did not collect data for products with no cartoons on packaging and so therefore we cannot include a reference group. But this is a valid point, and one that we will consider for future research. I have included a sentence within the limitations section (line 278).

3. Lines 138-139: Instead of imputing data from food composition databases (I guess) I would suggest to directly classify those products as non HFSS and green in all traffic lights.

RESPONSE: Of the 9 products without nutrition information on pack, 7 were fruits, vegetables and water. Nutrition information was obtained from McCance and Widdowson’s ‘The Composition of Foods’ as these are unprocessed food and drink products. We feel it is therefore acceptable and also fair to use generally accepted figures for these products, instead of making general assumptions on their fat, saturated fat, sugar and salt content and assuming they are all green on front of pack. Apples for example, contain 11.8g sugar per 100g, which would actually result in an amber traffic light on front of pack. The other 2 products were of processed foods; a chocolate and a jelly product. The nutrition composition of processed foods are highly variable as they depend on manufacturer and ingredients, and therefore we do not believe it would be accurate or fair to provide average figures for these two products, hence why they were excluded from the analysis.

4. Lines 145: Please remove the word healthiness... In fact, you need to revisit the manuscript and remove any comment suggesting that a nutrient profiling model classifies a food as healthy or unhealthy.

RESPONSE: The reviewer makes a valid point that the term ‘healthy vs unhealthy’ is indeed complex, and therefore we have tried to address this throughout the manuscript, amending where applicable the use of the term ‘unhealthy’. Nutrient profiling is used as a tool to help differentiate between foods and drinks that are more likely to be part of a healthy diet from those that are less likely (notably those foods that may contribute to excess consumption of energy, saturated fats, trans fats, sugar or salt).

5. Lines 152: remove comment on unhealthy... 

RESPONSE: We have also amended the manuscript to refrain from comments within the results section (line 166).

6. The sample for fruits and vegetables is too small to be analysed alone. Incorporate it to the previous sections licenced and non licensed characters with a comment on how many of the products in each analysis were fruits & vegetables.

RESPONSE: We have amended the manuscript to reflect this, by merging 3.2 and 3.3 within the results section (line 176). No statistical analysis was carried out for this.

7. Lines 191-196: You attempt a comparison with the previous survey but it is not possible for the reader to make the comparison. 

RESPONSE: With regards to a previous study 19 years ago (line 209), the research in question has been included to demonstrate that the use of cartoon animations on pack has been used as a marketing strategy by the food industry for many years. It is not our intention to make direct comparisons between our study and the one carried out in 2000, and we have amended the paragraph slightly to make this clearer (line 214). It would be difficult to make direct comparisons for a number of reasons: the studies did not follow the same methodology and inclusion/exclusion criteria, the sample size is different (358 in 2000 vs 534 in this research), and the criteria for assessing whether a product was high in fat, saturated fat, sugars and salt were different in the research referenced within this section.

Round 2

Reviewer 1 Report

The authors have roundly rejected the idea that there is any controversy relating to the issue of either saturated fat or salt intake.

Within their own view point, the article hangs together, however, I do not support its publication as I believe it is not scientifically sound to not at least acknowledge the controversy about reducing saturated fat and salt as useful public health interventions. 

So for me, who is not convinced about the utility of reducing saturated fat and salt, the manuscript is not appealing. However, I can see that for others, who follow this approach, it may be of interest.

The figure was improved in response to my comments.